# Predictive Image Regression for Longitudinal Studies with Missing Data

**Sharmin Pathan**
Department of Computer Science
University of Georgia

**Yi Hong**
Department of Computer Science
University of Georgia

## Abstract

In this paper, we propose a predictive regression model for longitudinal images with missing data based on large deformation diffeomorphic metric mapping (LDDMM) and deep neural networks. Instead of directly predicting image scans, our model predicts a vector momentum sequence associated with a baseline image. This momentum sequence parameterizes the original image sequence in the LDDMM framework and lies in the tangent space of the baseline image, which is Euclidean. A recurrent network with long term-short memory (LSTM) units encodes the time-varying changes in the vector-momentum sequence, and a convolutional neural network (CNN) encodes the baseline image of the vector momenta. Features extracted by the LSTM and CNN are fed into a decoder network to reconstruct the vector momentum sequence, which is used for the image sequence prediction by deforming the baseline image with LDDMM shooting. To handle the missing images at some time points, we adopt a binary mask to ignore their reconstructions in the loss calculation. We evaluate our model on synthetically generated images and the brain MRIs from the OASIS dataset. Experimental results demonstrate the promising predictions of the spatiotemporal changes in both datasets, irrespective of large or subtle changes in longitudinal image sequences.

## 1 Introduction

Since the last decade, longitudinal images are increasingly available for studying brain development and degeneration, disease progression, and aging problems. For instance, to understand the evolution of longitudinal data like brain scans over time, image regression [1] is a commonly-used technique to capture underlying spatialtemporal changes. This regression model estimates images as a function of associated variables like age under the framework of Large Deformation Diffeomorphic Metric Mapping (LDDMM) [2]. The following-up works aim at capturing non-linear changes with polynomial or spline regression [3, 4], modeling hierarchical changes at subject- and group-levels separately [5], or improving computational efficiency by introducing model approximations [6, 7]. These methods summarize the time-varying changes of a population, which is parameterized by the initial conditions of the captured smooth trajectory, e.g., the initial image and its associated initial momentum or velocity. To leverage this summarized trajectory and predict follow-up image scans for a specific subject, we need parallel transport techniques [8] to transport the estimated initial momentum or velocity from its corresponding initial image to the image scan of the target subject. However, parallel transport in image space is non-trivial, which is still under development. Another choice is image regression based on kernel methods [9, 10]. Typically, these approaches do not provide an explicit model that extracts parameters for further statistical analysis. Also, their prediction procedure highly depends on the training data, which is not efficient for a large dataset.

Recent advances and success in deep neural networks (DNN) [11] provide an alternative strategy to study longitudinal image populations. Several convolutional neural networks (CNN) and recurrent

1st Conference on Medical Imaging with Deep Learning (MIDL 2018), Amsterdam, The Netherlands.

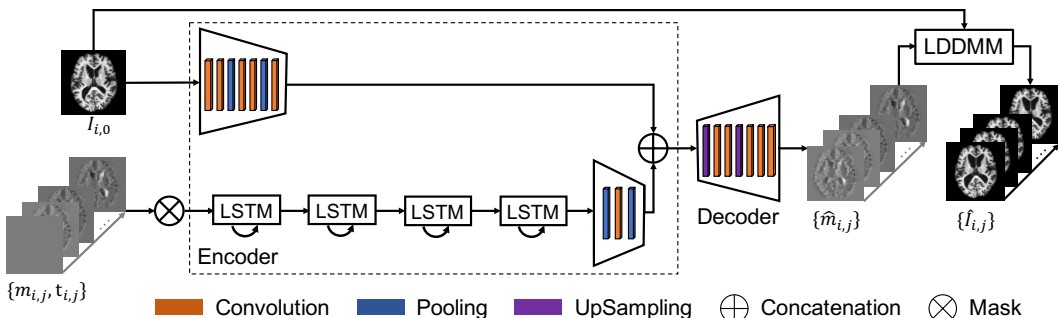

Figure 1: Architecture of our predictive image regression network. The baseline image $I_{i,0}$ of a subject $i$ passes through a CNN image encoder to extract its features, which are concatenated with features of a vector momentum sequence $\{m_{i,j}, t_{i,j}\}$ extracted by an LSTM encoder. The binary mask is used to ignore missing data at some time points. The concatenated feature maps are the input of a CNN decoder to reconstruct the vector momentum sequence $\{\hat{m}_{i,j}\}$, which is used to deform the image $\{I_{i,0}\}$ and predict the image sequence $\{\hat{I}_{i,j}\}$ in the LDDMM framework.

neural networks (RNN) [12, 13] have been proposed to predict the next frame of a video, without requiring the computation of complex regression models. However, because they treat images as a collection of pixel intensities without understanding their underlying geometrical structures, these data-driven methods based on DNNs often suffer a problem of blurry image predictions. Furthermore, different from video frame prediction, our task of predicting medical image scans aims to deal with longitudinal data collected at different time points with varying time intervals. In practice, subjects have image scans at different ages, and each subject may not scan regularly. As a result, we have the missing data issue at some time points, which does not often exist in video prediction.

In this paper, we address the problem of predicting follow-up image scans of a specific subject with a baseline scan as input, using a deep neural network learned from longitudinal data with missing scans. To learn a growth trend for a specific subject from a population and to overcome DNNs' blurry prediction issue for the follow-up scans, we integrate image registration techniques used in image regression models with a mixed CNN and RNN architecture, as shown in Fig. 1. Instead of directly working on input images to predict image sequence of a subject, our model predicts a sequence of vector momenta [14] for a baseline image input. These vector momenta are associated with their baseline image and parameterize deformation mappings between images under the LDDMM framework. We use the predicted vector momenta to deform the baseline image to different time points and generate the corresponding image sequence. We train our predictive image regression network by using the first image of a subject and its associated vector momentum sequence for each image scan, which is generated using LDDMM image registration. To handle the missing data, we introduce a binary mask into the training procedure, which ignores the loss calculation for predictions at missing time points. During the prediction procedure, given an individual with one image scan, our model can predict how it changes over time according to the learned population trend.

The most related work to ours is the fast image regression in [15], which uses pairwise fast image registrations [16] in a simplified image regression model [6] to summarize regression trajectories. The model proposed in [6] uses a distance approximation for measuring image differences, which assumes only small deformations existing among images. To relax this assumption, our model fully leverages longitudinal data and learns the deformations from time series data. We adopt an RNN composed of Long Short-Term Memory (LSTM) units, which is well suited for time-series prediction problems. We evaluate our model on both synthetic and real datasets. The experimental results demonstrate the effectiveness of our proposed model by capturing designed changes in the synthetic data and enlarging brain ventricles in the longitudinal OASIS dataset [17].

## 2   Predictive Network for Image Regression

In this section, we present the architecture of our predictive image regression network (Fig. 1) in detail. Assume we have a population of images collected from $N$ subjects and each subject $i$ has a varying

number ($P_i$) of images ($\{I_{i,j}\}_{j=0}^{P_i-1}$) scanned at different time points ($\{t_{i,j}\}_{j=0}^{P_i-1}$). The objective of image regression is to uncover the relationship between images $\{I_{i,j}\}$ and their associated variable $\{t_{i,j}\}$. Instead of directly performing regression on images, we leverage LDDMM and geodesic shooting with vector momentum [14, 18] to convert the longitudinal images of a subject into an initial image $I_{i,0}$ and a sequence of associated momenta $\{m_{i,j}\}_{j=0}^{P_i-1}$ (Section 2.1). The relation of the initial image and its momentum sequence to their associated variables, like age, will be learned through training a deep neural network (Section 2.2). The predicted momentum sequence for an input image can shoot it forward to generate the corresponding image sequence (Section 2.3).

## 2.1 LDDMM and Momentum Generation

Before studying image time series, we first need to establish image mappings, i.e., the correspondences between images. The LDDMM framework [2] provides a solution that estimates maps of diffeomorphisms (smooth mapping and smooth inverse mapping) to deform one image to another. Specifically, given a source image $\mathcal{I}_0$ and a target image $\mathcal{I}_1$, the LDDMM estimates a diffeomorphic mapping between them by minimizing the following energy:

$$E(v) = \int_0^1 \|v\|_L^2 dt + \frac{1}{\sigma^2}\|\mathcal{I}_0 \circ \Phi^{-1}(1) - \mathcal{I}_1\|_2^2,$$
$$s.t. \quad \frac{d\Phi}{dt} = v \circ \Phi, \ \Phi^{-1}(0) = Id.$$

Here, $v$ is a spatiotemporal velocity field, $L$ is a differential operator on the velocity field to enforce its smoothness, e.g., $L = -\alpha\nabla^2 - \beta\nabla(\nabla\cdot) + \gamma, \sigma > 0$ is a constant to balance the first regularization term and the second image matching term in the above equation, $\Phi$ is the diffeomorphic mapping, and $Id$ is the identity map. This formulation can be solved using the shooting strategy [18] and the image registration from $\mathcal{I}_0$ to $\mathcal{I}_1$ can be parametrized by a initial vector momentum $m_0$. Here, the $m_0$ is the dual of the velocity field [14], that is, $m = Lv$, which is associated with its initial image $\mathcal{I}_0$.

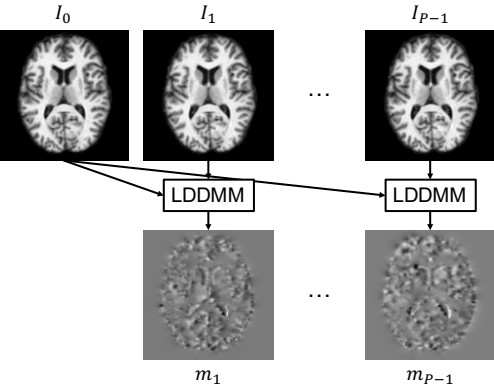

Figure 2: Generation of a vector momentum sequence using LDDMM. The baseline image ($\mathcal{I}_0$) is paired with every other image in the sequence ($\mathcal{I}_1, \mathcal{I}_2, ..., \mathcal{I}_{P-1}$). The registration of the baseline image to itself is not shown here.

Given a sequence of longitudinal images from a subject $i$, we select the first scan $I_{i,0}$ as the baseline image and compute the initial vector $m_{i,j}$ for each scan $I_{i,j}, (j = 0, \cdots, P_i - 1)$, by registering $I_{i,0}$ to $I_{i,j}$ using LDDMM, as shown in Fig.2. Note that the registration of the baseline image to itself is not shown in Fig. 2, since the resulting vector momentum is zero; but this zero momentum serves as a starting point for the recurrent network training and prediction in Section 2.2. Specifically, each vector momentum has $x$, $y$, and $z$ components and the $z$ dimension has zero momentum for a 2D image. As a result, we represent the image sequence of a subject as one image and its associate vector momentum sequence. Each vector momentum inherits the associated variable of its corresponding image. That is, we have a population of data represented as $\{I_{i,0}, \{m_{i,j}, t_{i,j}\}_{j=0}^{P_i-1}\}_{i=0}^{N-1}$ by using LDDMM and geodesic shooting.

## 2.2 Predictive Regression Network

The new data representation obtained in Section 2.1 brings the study of longitudinal images from the manifold of diffeomorphisms to the tangent space of the first scan, which has a Euclidean structure. Since the vector momenta are in a Euclidean space, their relationships to the associated variable, i.e., the age, can be learned using an RNN, which is designed to handle Euclidean time series data. Meanwhile, because the vector momenta are associated with their first image scan, we also include this baseline image in our network when learning features from momentum sequences. To predict a momentum sequence for image generation, we design an encoder-and-decoder network as shown in Fig. 1. This network has three components, i.e., a CNN image encoder to extract features from the

baseline image, an LSTM vector momentum encoder to handle the vector momentum sequences, and a CNN vector momentum decoder to reconstructs the vector momenta, which will be used to shoot the baseline image forward and generate follow-up image scans.

**CNN Image Encoder.**   This pathway in the network accepts the baseline image of a subject and extracts its image features that will be used for predicting the corresponding vector momenta. In particular, this feature extractor is a series of convolutional and pooling layers to learn hierarchical features from the first image scan $I_{i,0}$ of a subject $i$. The initial pair of convolutional layers have 32 filters and the filter number in the second pair increases to 64. A max pooling layer and a dropout [19] follow after every pair of convolutional units. The last convolutional layer in this CNN branch has 128 convolutional filters. Throughout the network, we use PReLU activation function [20], convolutional units with a $3 \times 3$ kernel size, and a max pooling size of $2 \times 2$. We choose the PReLU activation function because of negative values in the vector momenta and keep it consistent here. The extracted features by this branch are concatenated with those extracted from the LSTM vector momentum encoder and fed to the CNN vector momentum decoder for generating a momentum sequence.

**LSTM Vector Momentum Encoder.**   The second pathway in the network aims to learn time-varying changes from the vector momentum sequences $\{m_{i,j}, t_{i,j}\}$ generated by LDDMM and geodesic shooting. Here, a recurrent network with LSTM units learns the relationship between the vector momentum and its associated variable, the age. Since the changes start from the baseline image at $t_{i,0}$, we consider the relative age of images and adjust the independent variable as the age difference in years to the baseline image, that is, $\Delta t_{i,j} = t_{i,j} - t_{i,0}$. As a result, this branch learns the vector momentum as a function of the relative age, instead of the absolute age. Since we have a limited number of LSTM units, the maximum age difference accepted by this LSTM encoder is four years in our experiments. That is, we consider longitudinal images collected within five years, and the age difference between adjacent LSTM units is one year. We choose the number of LSTM units according to the OASIS dataset (Section 3), and it may be different for another longitudinal dataset.

In practice, it is very likely that a subject misses image scans in one or more follow-up years, which happens in the OASIS dataset. To deal with the missing data, we add a masking layer after the input layer to mask the missing inputs in the sequence. In particular, the mask layer has a time-distributed structure, which is an array of zeros for a missing time point while an array of all ones, otherwise. As a result, the predicted momenta for those missing time points will not be considered in the objective function calculation. This simple strategy allows us to handle a sequence prediction with one or multiple missing time points.

It is worth to mention that we use the convolutional LSTM [21] in our network. It has an LSTM architecture combined with CNN, which is specifically designed for sequence prediction problems with spatial inputs like images or videos. In particular, the convolutional LSTM includes convolutional layers to extract features from the input data and the LSTM units to support the sequence prediction. In this LSTM encoder, a total of four convolutional LSTM layers follow the masking layer. Among the four convolutional LSTM units, the first two units have 32 filters, which is then doubled to 64 in the next pair. A batch normalization layer is added after every convolutional LSTM layer. The output feature maps of the LSTM units have the same spatial size as the input momenta. To reduce the size of features maps and keep consistent with the CNN image encoder, after the LSTM units we use a max pooling layer, a convolutional layer with 128 filters, and another max pooling layer. The two pooling layers result in feature maps with the same size of the CNN image encoder for concatenation. The features learned from this network are then forwarded to the decoder network.

**CNN Vector Momentum Decoder.**   The decoder network aims to reconstruct a vector momentum sequence as the prediction $\{\hat{m}_{i,j}\}$ using the features learned from the baseline image and its associated momentum. To achieve this, we use an inverted version of the CNN image encoder architecture for the decoder. In particular, this network consists of an up-sampling layer (up-sampled by 2) and a pair of convolutional layers with 64 filters. These layers are followed by another up-sampling layer and another pair of convolutional layers with 32 filters. In this way, the network can reconstruct the momenta to the original input size. This decoder takes the features maps learned from the encoder network as input and predicts the next vector momentum in the sequence. We append this newly-predicted momentum to the previously-predicted sequence, which is fed to the LSTM units again to predict momentum at a further time step. This recursive call continues until we complete the

---
**Algorithm 1** Workflow of Our Predictive Network for Image Regression
---
 1: **procedure** PREPROCESSING
 2:     Compute the vector momentum sequence $\{m_{i,0}, m_{i,1}, ..., m_{i,P_i-1}\}$ of each subject $i$ using LDDMM and geodesic shooting for image pairs $\mathcal{I}_{i,0} \to \mathcal{I}_{i,0}$, $\mathcal{I}_{i,0} \to \mathcal{I}_{i,1}$, $\mathcal{I}_{i,0} \to \mathcal{I}_{i,2}$, ..., $\mathcal{I}_{i,0} \to \mathcal{I}_{i,P_i-1}$, as shown in Fig. 2.
 3:     Construct vector momentum sub-sequences, for instance, $m_{i,0}$ has a target $m_{i,1}$, $m_{i,0}$ and $m_{i,1}$ together have a target $m_{i,2}$, and so on.
 4:     Save the initial images $\{\mathcal{I}_{i,0}\}$ with their associated vector momentum sequences $\{m_{i,j}, t_{i,j}\}$.

 5: **procedure** TRAINING
 6:     Pre-train the CNN vector momentum decoder using computed vector momenta from step 2. This is achieved by training an autoencoder to reconstruct the vector momenta.
 7:     Train the CNN image encoder on initial images of all subjects $\{\mathcal{I}_{i,0}\}$ through training an autoencoder to reconstruct the images.
 8:     Train the LSTM encoder on vector momentum sequences constructed from step 3 and use binary masks to handle missing time points.
 9:     Merge the features extracted from steps 7 and 8 and feed them to the decoder.
10:     Fine-tune the weights of the decoder by training it over the merged features from the two encoder branches.

11: **procedure** PREDICTION
12:     Extract features from an input image $\mathcal{I}_0$ using the CNN image encoder.
13:     Set $m_0$ to be zero.
14:     Feed $m_0$ to the LSTM vector momentum encoder.
15:     Feed extracted features from the input image and initial momentum $m_0$ to the decoder.
16:     The decoder predicts the next vector momentum $\hat{m}_1$.
17:     Append $\hat{m}_1$ and form a momentum sequence with $m_0$.
18:     Extract features from this newly-formed sequence by passing it as input to the LSTM network to predict $\hat{m}_2$.
19:     Repeat steps 17 and 18 to predict $\hat{m}_3, \hat{m}_4, ...,$ until $\hat{m}_{s-1}$ is predicted. Here, $s$ is the maximum sequence length that the LSTM network can handle.
20:     Apply the predicted vector momentum sequence to $\mathcal{I}_0$ with LDDMM shooting to generate the sequence of the follow-up images $\hat{\mathcal{I}}_1, \hat{\mathcal{I}}_2, ..., \hat{\mathcal{I}}_{s-1}$.
21:     If needed, treat $\hat{\mathcal{I}}_{s-1}$ as $\mathcal{I}_0$ and $\hat{m}_{s-1}$ as $m_0$ to continue the prediction, repeating steps 14-20.
---

prediction of the last momentum in the sequence. The vector momentum output has $x$, $y$, and $z$ three components, which is generated by a convolutional layer with three filters.

## 2.3 Network Training and Prediction

In our regression model, we use the mean squared error (MSE) as the loss function for either pre-trained or trained networks. Algorithm 1 depicts the workflow of our predictive network for image regression. It includes the preprocessing procedure discussed in Section 2.1, the training procedure for the network components discussed in Section 2.2, and the prediction procedure for generating the image sequence. Although the LSTM sub-network accepts a fixed number of vector momenta in the sequence, we can predict more momenta and generate more images by taking the last prediction of the vector momentum and its corresponding image as the initial momentum and the initial image to continue the prediction procedure.

## 3 Experiments

**Datasets.** We evaluate the prediction performance of our predictive network on both synthetic and real datasets. The synthetic data is a set of binary images of concentric circular rings like the bull eyes shown in Fig. 3. The radii of the concentric rings change with a constant rate but having a small Gaussian noise. It has 52 2D image sequences with image size of $64 \times 64$, and each sequence has five time points. In this synthetic dataset, 40 momentum sequences are randomly selected for

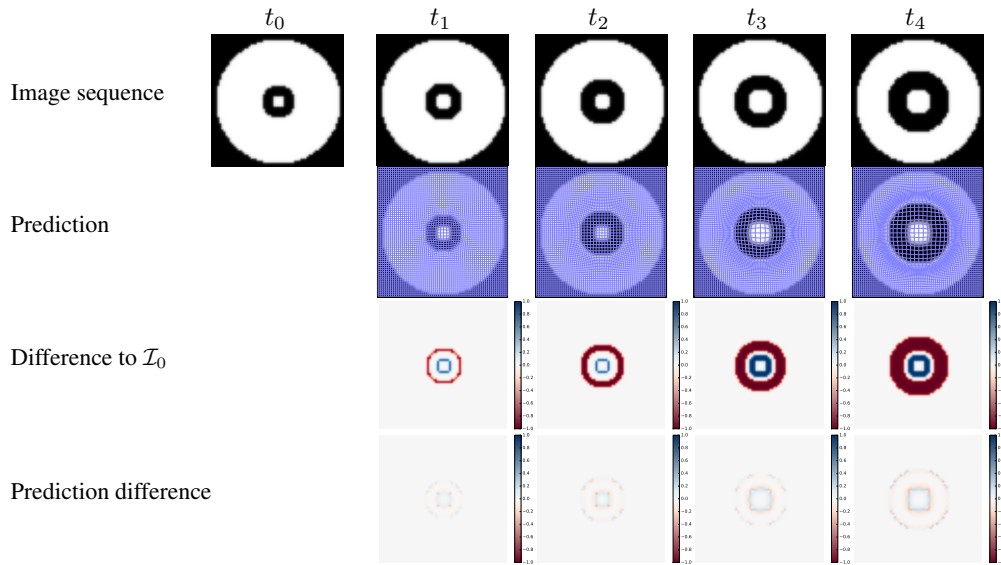

Figure 3: Prediction results for one sample image from the synthetic test set. The first row shows the original image sequence, followed by our predicted sequence with deformation maps (the blue grids). The third row shows the image difference between the first image $\mathcal{I}_0$ and its follow-up images. The last row demonstrates the image difference between our prediction results and the corresponding ground-truth images. Best viewed in color.

| Synthetic data (1e-4) | Non-demented Group (1e-4) | Demented Group (1e-4) |
|---|---|---|
| $8.1184 \pm 5.3171$ | $5.8616 \pm 1.0703$ | $6.1105 \pm 1.9793$ |

Table 1: The mean and standard deviation of mean squared errors over all images in each test group.

training, reserving eight sequences for validation and four sequences for testing (total 16 images for prediction). The estimated vector momenta using LDDMM is of the form $64 \times 64 \times 3$, and the third channel, i.e., the $z$ component, is zero. We normalize the image intensity within 0 and 1 for all images. In this dataset, we do not consider the missing data problem.

The real dataset includes 2D image slices of brain MRIs from the OASIS database. We have 136 subjects aged from 60 to 98, and each individual was scanned at 2-5 time points with the same resampled resolution $128 \times 128$ and the voxel size of $1.25 \times 1.25$ mm$^3$. All images were preprocessed by down-sampling, skull-stripping, intensity normalization to the range [0,1], and co-registration with affine transformations. The vector momenta generated between pairs of these images are 128x128x3. We evaluate our model on the non-demented and demented groups separately. In particular, the non-demented group has 72 image sequences, 58 used for training, 7 for validation, and 7 for testing (total 12 images for prediction); and the demented group has 64 image sequences, 52 of them are used for training, 6 for validation, and 6 for testing (total 9 images for prediction).

**Experimental Settings.** In LDDMM, we set the parameters for the $L$ operator to $[\alpha, \beta, \gamma] = [0.01, 0.01, 0.001]$ and $\sigma$ to 0.2. In the network, we use Adam optimizer [22] and a dropout rate of 0.5. In the training procedure, we train the CNN image encoder for 250 epochs, pre-train the CNN vector momentum decoder for 1500 epochs, and train the LSTM vector momentum encoder and fine-tune the CNN vector momentum decoder for 500 epochs.

**Experimental Results.** Figure 3 demonstrates the prediction results of one test sample in the synthetic dataset. The first two rows show the original image sequence and our predicted one, and the last two rows show the image difference. In particular, we compute the image difference between each predicted image and its corresponding image in the sequence, as shown in the last row of Fig. 3, and compare it with the image changes relative to the first image in the sequence, as shown in the third row of Fig. 3. The dramatically reduced image difference indicates our prediction is promising. In the second row of Fig. 3, the deformation maps overlapped on each image also validate that our predictive

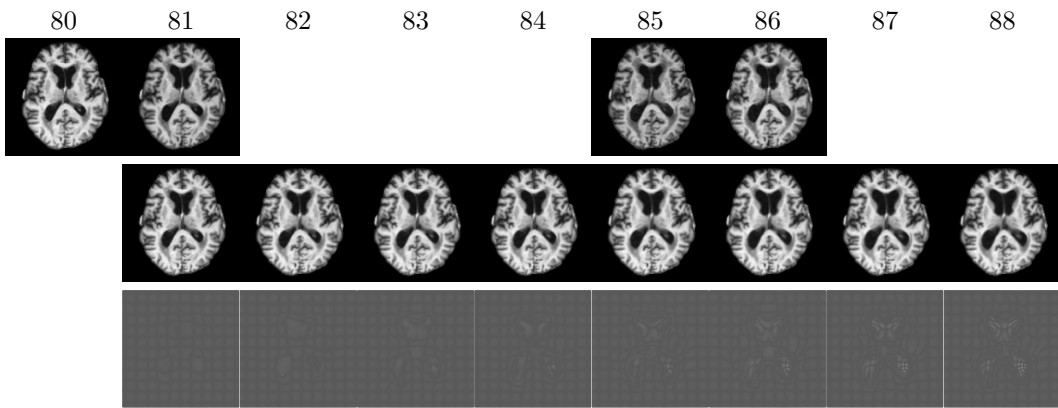

Figure 4: Prediction results of one sample subject from the non-demented test group of the OASIS dataset. The first row shows image scans of the subject at 80, 81, 85, and 86 years, and images at the other time points are missing. The second row shows our predicted image scans for this subject with the first scan at 80 years as the baseline image. The last row shows the corresponding deformation map for each predicted image. Best viewed in color and zoom in.

model correctly captures the expanding changes designed in the synthetic data. We compute the MSE between an image and its prediction and estimate the mean and standard deviation for all predicted images in the test set of the synthetic data. As reported in Table 1, the mean image difference is 8.1184e-4±5.3171e-4. Note that, since the image intensity is within [0, 1], the maximum possible value for the mean image difference is 1.

Figure 4 shows the prediction results of our model for one subject sample from the non-demented test group in the OASIS dataset. This subject has MRI scans at 80, 81, 85, and 86 years old, and there are missing images at multiple time points. We predict one image scan per year from 80 years to 88 years, including those missing ones, as shown in the second row of Fig. 4. Since the brain changes are quite subtle (see the first row of Fig. 5), we also plot the deformation map $\Phi$ at each time point, as shown in the last row of Fig. 4. These deformation maps show the estimated changes in the brain MRIs. As we can see, the grids are expanding, especially around the brain ventricle region. This expanding ventricle indicates our model captures the degeneration process of the ventricle in the brain, i.e., an enlarging ventricle. Figure 5 shows the image difference between predicted images and their corresponding image scans in the second row. Compared to the first row that shows image difference of follow-up scans with respect to the first one, the prediction difference is relatively smaller, especially around the ventricle region. Table 1 reports the means and standard deviations of the prediction difference for all images in the non-demented and demented groups, which are 5.8616e-4±1.0703e-4 and 6.1105e-4±1.9793e-4, respectively.

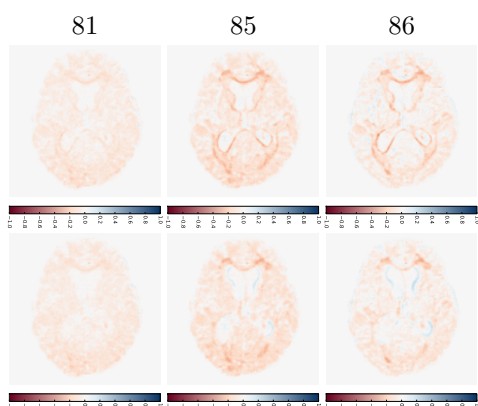

Figure 5: Difference plots for image scans and their predictions in Fig. 4. The first row shows the image difference of image scans at 81, 85, and 86 years compared to the first scan of the subject at 80 years. The second row shows the difference between our predictions and image scans collected at the same years. Best viewed in color.

Apart from the subject-specific image prediction, our model can also estimate a group trajectory by predicting forward and backward image sequences for the atlas (the mean image) built for that group. Figure 6 demonstrates the mean trajectory estimated for the demented group. We first estimate the mean image using the unbiased atlas building algorithm [23]. This atlas is our baseline input, and we predict vector momenta forward to generate future image scans. By using the negative vector

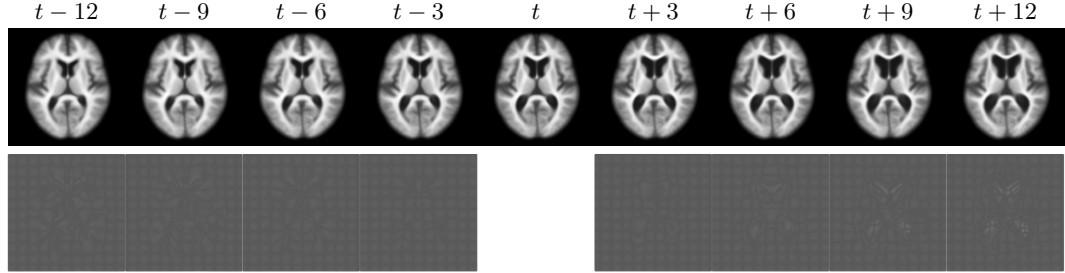

Figure 6: Forward and backward predictions of the image sequences (top row) for the atlas (at time t) of the demented group in the OASIS dataset. The second row shows their corresponding deformation maps. Best viewed in color and zoom in.

momenta, we shoot the atlas backward and generate previous image scans. From the predicted images within 25 years (every three years shown in Fig. 6), we can recognize the brain changes, in particular, the enlarging ventricle over the years. The deformation maps shown in the second row are generated starting from the atlas in the middle. Therefore, they show the expanding grids in the forward sub-sequence and the compressed grids in the backward sub-sequence.

## 4 Discussion and Conclusions

In this paper, we proposed a novel approach to predict time-varying medical image scans by integrating topologically-preserving image registration model (LDDMM) with deep neural networks. This model not only inherits the good properties from LDDMM that guarantee a sharp image prediction but also leverages the deep learning merits of learning from data, without the need of parallel transport for specializing the prediction for a specific subject. We implemented our predictive model to support the 2D image prediction; however, it can be straightforwardly extended for predicting 3D images. One possible challenge in 3D implementation is a shortage of GPU memory, which could be solved by using 3D image patches instead of the whole 3D volume.

One limitation of our method lies in the LSTM network, which mainly focuses on capturing linear changes in the image sequence, due to its shared weights among units of a layer at all time points. In the future work, we will consider of another type of recurrent neural networks or improve the current LSTM architecture to capture non-linear changes. In addition, the network cannot deal with missing correspondences among images, such as a tumor appearing or disappearing in brain images. This limitation is caused by the LDDMM used in this paper, which cannot handle image registrations with missing correspondences. To address this issue, we could replace the pairwise image registrations with image metamorphosis [24], which was developed under the LDDMM framework.

## Acknowledgments

This research was supported by a 2018 UGA Faculty Research Grant. The Titan X Pascal used for this research was donated by the NVIDIA Corporation.

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
