# OpenReview forum: "Predictive Image Regression for Longitudinal Studies with Missing Data"
_MIDL.amsterdam/2018/Conference — MIDL 2018 Poster_

### Review · AnonReviewer2 · 2018-05-08
**interesting first step and proof of principle**

**Rating:** 4
**Confidence:** 2

**Review:**

An LSTM network learns output of LDDMM. The architecture uses CNN encoded baseline images and the LSTM output in a joint decoder to predict the forward and backward trajectory of longitudinal brain scans.

Pro: To the best of my knowledge, the idea is novel to encode longitudinal deformation fields in longitudinal brain aging data sets. This is an interesting proof of principle and results are promising.

Con: The pipeline for training and applying the algorithm is a pipeline, i.e., it is a pipeline and far away from end to end approaches. Predicting a trajectory from a single brain scan remains a somewhat arbitrary problem and it would be interesting to see if the method generalizes to predicting trajectories from short sequences of two, three, or four longitudinal images.




**Special Issue:**

Yes

---

### Review · AnonReviewer3 · 2018-05-09
**Interesting idea that requires a more convincing validation**

**Rating:** 3
**Confidence:** 2

**Review:**

In this paper, the authors propose a framework for the prediction of follow-up image scans from a baseline input image, using a deep neural network trained with longitudinal data with missing scans. Instead of directly predicting the resulting image, the authors train the network to generate the vector momentum that transform/register the baseline image to the corresponding time-point. The proposed architecture is evaluated using synthetic and real 2D MR brain scans from the OASIS database, using healthy and demented cases. The idea to combine CNN with LSTM to predict deformation fields is not completely novel, however, its application to estimate LDDMM-based vector momentum on brain MRIs and predict their evolution over time is very interesting. However, the performance of the proposed method on demented cases is not clearly described and further experiments would need to be done in order to fully evaluate its clinical potential.
In general, the paper is well written and is easy to follow. However, some additional technical details regarding the architecture implementation and the training process (e.g., loss function, optimizer, number of epochs, computational cost) need to be provided.
Strengths:
-	The idea of using deep learning to predict the progression of longitudinal studies is interesting.
-	To redefine the image prediction problem as a simple regression task.
Weaknesses:
-	The validation of the proposed architecture is very limited. There is no graphical example of how it performs with pathological cases.
-	To use the age difference as independent variable can be controversial. That means when predicting the next year scan of a patient, it wouldn’t matter if the patient is 60 or 90 years old.
-	There are many technical details that need to be clarified.

Additional comments:

	Please, complete Fig. 1 including the architecture details (i.e., filter size, number of filters, etc).

	They use PRELU as activation function. Unlike RELU, PRELU has a non-zero slope in the negative size, necessary to deal with the negative values in the vector momenta. Normally, PRELU has an asymmetric slope, where the slope of the negative size is an hyper parameter. How is this parameter chosen? How does it affect the prediction of the vector momenta (particularly the negative values)?

	The data were evaluated on non-demented and demented groups separately. How would the proposed architecture behave if combining both images simultaneously? Did the network hallucinate dementia features for healthy cases? Did it underestimate the degeneration effect in demented patients? This is an interesting experiments the authors should perform in order to evaluate the potential of the method for early detection of dementia, or predicting tool.

	It is hard to evaluate visually the prediction results presented in Fig. 4 (non-demented case). The deformation maps show some evolution over time, but the resulting images seem unchanged. The main observable effect is located on the lateral ventricles. Is this deformation patter specific of this patient or is it a systematic deformation observed in all the healthy cases?

	There is no figure showing the performance of the architecture on demented cases. It is expected that the changes over time to be more severe, and even patient-dependent. Was the network able to effectively predict the evolution of specific patients with dementia? This is crucial in order to prove the clinical potential of the proposed architecture.

	The authors should provide a detailed description of the training process: Number of epoch, optimizer, etc, as well as details of the computational cost.

	When predicting the images for t+2, the model initially predict the vector momentum for t+1, using t=0 and t+1 as predictors or t+2. How does the prediction change when using the estimated t+1 or the groundtruth for t+1? Also, how does the prediction accuracy evolve with time (i.e., as we move further from the original time point)?

	Please, describe the loss functions used for training. As described in Algorithm 1, each sub-part of the network is pre-trained separately, but no information regarding the corresponding loss function is mentioned. Also, is there any constraint in the loss function when estimating the vector momentum that guarantee the smoothness of the estimated vectors?


**Special Issue:**

Yes

---

### Review · AnonReviewer1 · 2018-05-09
**Predictive Image Regression for Longitudinal Studies with Missing Data**

**Rating:** 3
**Confidence:** 2

**Review:**

The paper presents a method for prediction of missing images in longitudinal images of 3D MRI scans using RNNs and CNNs. The idea is to use a baseline scan to predict subsequent scan. For this purpose, vector momenta are used, as well as convolutional networks and 3D convolutional LSTM. Validation on a synthetic and a real dataset of MRI images is performed.

The work presented in this paper relies on previous technology, which is integrated with deep learning techniques.
Although this approach is valid, additional details on previous technology used here should be provided, in order to ease the readability and clarity of the paper.
Moreover, the proposed method is not compared with other methods, and mostly visual results (and some quantitative results, in terms of difference) are reported.
Discussion on whether predicted images can have a plausible biological interpretation would have been a plus.

Additional comments:
* A 0/1 mask is mentioned in Figure 1 and in the introduction, but it is not clear what it is and how it is used
* Geodesic shooting is mentioned and used, but not explained
* Convolutional LSTM are not largely used yet, a short explanation/introduction would be useful
* The proposed approach processes 2D data, but this information is not reported until the results section.


**Special Issue:**

No

---

### Decision · Program_Chairs · 2018-05-15
**Paper116 Acceptance Decision**

Poster